# Polygenic Risk Scores Associated with Tumor Immune Infiltration in Common Cancers

**DOI:** 10.3390/cancers14225571

**Published:** 2022-11-14

**Authors:** Jungyoon Choi, Jung Sun Kim, Hwa Jung Sung, Yu-Wei Chen, Zhishan Chen, Wanqing Wen, Xiao-ou Shu, Xingyi Guo

**Affiliations:** 1Division of Oncology/Hematology, Department of Internal Medicine, Korea University Ansan Hospital, Korea University College of Medicine, Ansan 15355, Korea; 2Division of Epidemiology, Department of Medicine, Vanderbilt Epidemiology Center, Vanderbilt-Ingram Cancer Center, Vanderbilt University School of Medicine, Nashville, TN 37203, USA; 3Department of Biomedical Informatics, Vanderbilt University School of Medicine, Nashville, TN 37203, USA

**Keywords:** cancer genetic susceptibility, genome-wide association study, polygenic risk score, tumor immune infiltration

## Abstract

**Simple Summary:**

Polygenic risk scores (PRSs) have been used to predict the risk of developing cancer. However, whether PRSs are associated with immune infiltration in solid tumors remains unclear. We constructed PRSs for common cancers of the breast, colorectum, lung, ovary, pancreas, and prostate using risk variants identified in previous genome-wide association studies, and comprehensively evaluated their associations with 139 immune traits previously estimated from The Cancer Genome Atlas. We identified 31 significant associations between PRSs and immune traits at a nominal (*p* < 0.05) level of significance. In the analyses stratified by stage for breast cancer, colorectal cancer, lung adenocarcinoma, and lung squamous cell carcinoma, we identified 65 significant associations, including 56 associations that were undetected by the overall analysis. Our findings provide novel insights into the role of genetic susceptibility in the immune responses underlying cancer development, prognosis, and the potential role of an early diagnostic or therapeutic targeting strategy.

**Abstract:**

It is largely unknown whether genetic susceptibility contributes to tumor immune infiltration in common cancers. We systematically investigated the association between polygenic risk scores (PRSs) and tumor immune infiltration in common cancers. First, we constructed a PRS for common cancers using the risk variants identified in previous genome-wide association studies. Then, we analyzed 139 immune traits predicted by previous studies by examining gene expression data in tumor tissues from The Cancer Genome Atlas (TCGA). We applied regression analyses to evaluate the associations between PRS and immune traits for each cancer overall and stratified by stage, including 2160 pathologically confirmed cases of breast, colorectal, lung, ovarian, pancreatic, and prostate cancers in the White population. At a nominal (*p* < 0.05) significance level, we identified 31 significant associations between PRS and immune traits. In the analyses stratified by stage for breast, colorectal, lung adenocarcinoma, and lung squamous cell carcinoma, we identified 65 significant associations, including 56 associations that were undetected by the overall analysis. This study provides evidence for genetic risk factors affecting immune infiltration and provides novel insights into the role of genetic susceptibility in immune responses, underlying cancer development, prognosis, and the potential role of an early diagnostic or therapeutic targeting strategy.

## 1. Introduction

Genome-wide association studies (GWAS) have identified genetic risk variants in common cancers ranging from approximately 20 variants linked to lung cancer to 300 variants linked to breast cancer [1,2,3,4,5,6,7]. Although the effect of each GWAS-identified risk variant is small, the cumulative effect of individual risk variants, measured by a polygenic risk score (PRS), on cancer risk is substantial. PRSs have been used to predict the risk of cancer development [1,2,3,4,5,6,7]. However, the underlying mechanisms in individuals at high risk of cancer remain largely unknown.

The tumor microenvironment (TME) plays a critical role in cancer development and progression. This affects the clinical outcomes of patients [8,9,10,11,12,13,14]. The TME mainly includes tumor-infiltrating immune cells combined with stromal components, such as tumor-associated blood vessels, extracellular matrix, and connective tissue cells. Immune traits are defined as the phenotypic diversity of TME components. Accumulating evidence has shown that immune traits, especially those of tumor-infiltrating lymphocytes (TILs) in the TME, correlate with immune checkpoint inhibitor responses, different histological characteristics of cancer, disease-free survival, cancer-specific survival, and overall survival in several cancer types, including colorectal, ovarian, and breast cancers [15,16,17,18,19,20]. In addition, immune checkpoint therapy helps the adaptive immune system to interfere with immune escape caused by the sequential activation of immune checkpoints, such as those controlled by programmed cell death protein (PD-1), which is expressed on activated T cells. Immune responses are inhibited when the PD-1 protein interacts with its ligand, programmed death ligand 1 (PD-L1), and cytotoxic T lymphocyte-associated protein 4 (CTLA-4), which is an inhibitory checkpoint on T cells that prevents the uncontrolled expansion of activated T cells [16]. However, the mechanisms of the immune traits that affect these tumor responses and clinical outcomes are still unknown, and comprehensive studies based on large sample sizes are lacking.

Previous studies have shown that germline genetic variations contribute to differences in immune infiltration in solid cancers [2,5,6,21,22,23]. These findings highlight that genetic susceptibility may contribute to cancer risk by mediating innate and adaptive immune responses [22]. However, it is unknown whether PRSs built from genetic variants, known to affect cancer risk, can predict immune infiltration of the TME in solid cancers. In this study, we constructed PRSs for common cancers of the breast, colorectum, lung, ovary, pancreas, and prostate using risk variants identified from previous GWAS for each cancer. To estimate the abundance of tumor immune infiltration in these cancers, we comprehensively analyzed 139 immune traits, as determined based on previous immune analysis of gene expression data in bulk tumor tissues from The Cancer Genome Atlas (TCGA) [24]. We systematically evaluated the association between PRS and immune traits in these common cancers.

## 2. Materials and Methods

### 2.1. Data Sources

Clinical and gene expression data from TCGA projects were obtained from the Genomic Data Commons Data Portal (https://portal.gdc.cancer.gov, accessed on 1 January 2022) and the corresponding publications. Data access for this study was granted through the Database of Genotypes and Phenotypes (dbGap) Project #24541. For immune traits, we merged two data sources: the feature matrix in the study by Thorsson et al. (56 immune-related features selected) and the scores for 160 gene signatures in tumor samples (160 features, Scores_160_Signatures.tsv) across 9769 individuals [25]. This study was exempted from institutional review board approval and informed consent.

### 2.2. Genotype Data and Imputation

Genotyping data generated by the Affymetrix Genome-Wide Human Single Nucleotide Polymorphism 6.0 Array were downloaded for breast cancer, colorectal adenocarcinoma, lung adenocarcinoma, lung squamous cell carcinoma, ovarian serous cystadenocarcinoma, pancreatic adenocarcinoma, and prostate adenocarcinoma from the TCGA legacy archive. In addition, we used genotype data from the mixed populations of the 1000 Genomes Project Phase 3 and Minimac tool [26,27], as implemented in the Michigan Imputation Server. Only common genetic variants (minor allele frequency > 0.05) with a high imputation quality (R^2^ > 0.3) were included in the study. Variants were excluded if they had outliers for heterozygosity, were mapped to sex chromosomes, or had low call rates. Those corresponding to individuals of non-white origin based on principal component (PC) analysis using variants intersected in the 1000 Genomes Project Phase 3 were excluded from subsequent analyses. These filters were applied because the GWAS used to provide summary statistics are typically restricted to populations of European origin. Additionally, we included only microsatellite instability–stable (MSS) tumors, as determined using whole-exome sequencing data from a previous study [28].

### 2.3. Construction of PRSs

Previously reported genetic risk variants for the common cancers studied were selected by reviewing the recent literature on GWAS and focusing on studies with the largest sample sizes of individuals of European ancestry (Appendix A). We used the conventional genome-wide significance threshold (*p* < 5 × 10^−8^), and genetic variants showing an association with *p*-values at or below this threshold were included in this study. Of note, additional GWAS-identified risk variants reported from earlier studies could not reach the genome-wide threshold in the meta-analysis owing to a smaller sample size, as these variants were selected for replication in the different stages using extra samples in the earlier studies. Although there were very few such variants, we included them in our analysis. GWAS-identified risk variants on the X chromosome, and those reported exclusively in non-European populations were excluded. For variants in linkage disequilibrium (LD r^2^ ≥ 0.2) in European ancestry populations in the 1000 Genomes Project, only the variant with the lowest *p*-value was included. We selected 807 unique risk variants for the common cancers that were analyzed from previously reported GWAS data. Among them, 793 risk variants were available in the TCGA data, and PRSs were built for each cancer using the GWAS risk variants identified for that cancer. We calculated PRSs by computing the sum of risk alleles harbored by an individual, weighted by effect sizes estimated from the most recent GWAS [29].

### 2.4. Categorizing Immune Traits from Tumor Tissues

To estimate the abundance of tumor immune infiltration in the TME for each cancer type, we used 139 well-characterized immune traits reported by Sayaman et al. [22]. Among 216 gene expression data estimated in the TCGA immune analysis [30], Sayaman et al. selected the final 139 immune traits through a data filtering process for data with filters, such as redundancy, limited interpretability, and skewed distribution. We used the same “categories” term listed in the study by Sayaman et al. to describe the methodological origin of the immune trait measures: leukocyte subset enrichment score, leukocyte subset percentages (%), overall proportion, adaptive receptor, expression signature, and attractor metagene. We also used the term “module” to describe the grouping of immune traits that were generated, based on clustering by their Pearson correlation coefficients, and six groups of correlated traits were defined: lymphocyte infiltration, macrophage/monocyte, IFN response, TGF-β response, wound healing, T cell/cytotoxic, and unassigned [22]. The term “immune subtype” was used to describe the immune grouping of samples previously characterized by Thorsson et al.: wound healing (C1), IFN-γ dominant (C2), inflammatory (C3), lymphocyte depleted (C4), immunologically quiet (C5), and TGF-β dominant (C6) [30]. These terms are summarized in Appendix A.

### 2.5. Statistical Analysis

In this study, we only included participants of European ancestry from the TCGA dataset, based on two PCs for genetic ancestry for each cancer type, as the PRSs were derived using risk variants for eight cancers identified from GWAS conducted in this population. The overall study design is shown in Figure 1A. The outcome variables included immune traits as numerical and categorical variables.

To investigate the associations between PRSs for cancers and 139 immune traits for each cancer overall and stratified by stage, we used linear regression models adjusted for sex, histological subtype, stage, smoking history, the top five PCs for genetic ancestry, and the sequencing center (covariates differed by cancer type availability; Table 1). For immune subtypes, we used multinomial regression models adjusted for the same covariates. Odds ratios (ORs) and 95% confidence intervals (CIs), associated with each PRS, were estimated using regression models for each cancer type. Statistical inferences were based on two-sided tests, with a significance level of 0.05. All statistical analyses were conducted using the R software version 3.3.3 (R Project for Statistical Computing).

## 3. Results

### 3.1. Study Design

We included 2160 pathologically confirmed common cancer cases with unique patient records and matched the clinical and sequencing data for 670 breast, 178 colorectal, 61 ovarian, 125 pancreatic, and 333 prostate cancer cases, along with 359 lung adenocarcinoma and 334 lung squamous cell carcinoma cases (Table 1). We constructed PRSs using risk variants identified in a previous GWAS and analyzed the abundance of tumor infiltration for the previously identified 139 immune traits (Figure 1A). We systematically evaluated the association between PRSs and immune traits in each cancer type. We also stratified the patients by stage for breast cancer, colorectal cancer, lung adenocarcinoma, and lung squamous cell carcinoma.

### 3.2. Overall Associations between PRSs and Immune Traits

We identified 31 significant associations corresponding to 26 immune traits that were associated with PRSs after combining the results from these common cancers (nominal *p* < 0.05). Of these, 14 significant immune traits were observed in prostate cancer, 5 in colorectal cancer, 4 in breast cancer and lung squamous cell carcinoma, 2 (interleukin [IL]-13 score 21050467 [member of TGF-β response module] and gamma delta T cells (%)) in lung adenocarcinoma, and 1(Troester WoundSig 19887484 [member of TGF-β response module] and eosinophils [member of T-cell/cytotoxic module]) in ovarian and pancreatic cancers. Overall, we observed the highest proportion of significant associations between PRS for prostate cancer and immune traits (Figure 1B). More specifically, we observed positive associations between PRSs of prostate cancer and immune traits, except for resting dendritic cells (DCs) (%). Antitumoral effects were positively associated with PRSs for T helper 1 (Th1) cells, which were part of the T-cell/cytotoxic module (OR, 1.01; 95% CI, 1.003–1.01; *p* = 0.0003), plasma cells (%) [member of unassigned module] (1.01; 1.01–1.02; *p* = 0.0007), and MCD3 CD8 21214954, which was also part of the T-cell/cytotoxic module (1.10; 1.03–1.17; *p* = 0.0052). A pro-tumoral effect was inversely associated with PRSs in resting DCs (0.997; 0.994–0.999; *p* = 0.0109). However, pro-tumoral effects were positively associated with PRSs in the stromal fraction (%), which was part of the lymphocyte infiltration module (1.04; 1.01–1.07; *p* = 0.0194), and angiogenesis, which was part of the T-cell/cytotoxic module (1.01; 1.0002–1.02; *p* = 0.0463) (Figure 1C, Appendix A).

### 3.3. Associations by Stage between PRSs for Cancers and Immune Traits

We conducted association analyses stratified by stage (early stage: I/II; advanced stage: III/IV). Stage information and sample size were available for the four cancer types (breast, colorectal, lung adenocarcinoma, and lung squamous cell carcinoma) (Table 1).

We identified 65 significant associations corresponding to 61 immune traits that were associated with PRSs (nominal *p* < 0.05) (Figure 2). Compared with the results of the overall association analysis, 15 significant associations corresponding to 13 immune traits were commonly observed among the four cancer types (Figure 1C). Of these 65 associations, we observed 7 (early stage) vs. 26 (advanced stage) significant associations for breast cancer, 19 vs. 2 for colorectal cancer, 1 vs. 5 for lung adenocarcinoma, and 1 vs. 4 for lung squamous cell carcinoma. The most significant associations in the advanced stages of breast cancer were associated with IFN response modules, and the most significant associations in the early stages of colorectal cancer were associated with the lymphocyte infiltration module. Interestingly, in lung cancer, we observed two significant associations that were commonly detected in the early and advanced stages. For lung adenocarcinoma, a 0.1% and 0.2% increase in the T cell gamma delta (%) [module of unassigned module] showed positive associations with PRSs in the early (OR, 1.001; 95% CI, 1.0001–1.002; *p* = 0.0365) and advanced stages (1.002; 1.0002–1.003; *p* = 0.0353). For lung squamous cell carcinoma, a 6% and 10% increase in lymphocytes (%) [module of unassigned module] showed positive associations with PRSs in the early (1.06; 1.0–1.11; *p* = 0.0123) and advanced stages (1.10; 1.01–1.20; *p* = 0.0406), respectively. The remaining 61 associations were only significant at specific stages (Appendix A).

### 3.4. Overall Proportion of Associations for Immune Modules

After identifying 31 significant associations, we characterized them based on seven immune modules, which were classified by collapsing 139 immune traits. These associations were assigned to immune modules: unassigned (18), T cell/cytotoxic (7), lymphocyte infiltration (3), and TGF-β response modules (3). When each of these modules was evaluated using the PRSs for the cancer types, lymphocyte infiltration showed prominent associations with the PRS for colorectal cancer (66.7%, 2/3), and T cell/cytotoxic modules showed prominent associations with the PRS for prostate cancer (71.4%, 5/7) (Figure 3A).

Among the 65 significant additional associations, across the four cancer types by stage, we observed that these associations mostly occurred in the immune modules of lymphocyte infiltration (24), unassigned (16), IFN response (9), macrophage/monocyte (9), T cell/cytotoxic (5), TGF-β response (1), and wound healing (1). The proportion of cancer types in each immune module is shown in Figure 3B. The absolute number of associations according to immune modules and stages is shown in Figure 3C.

### 3.5. Putative Predictive Biomarkers for Clinical Practice Associated with Antitumor Immune Responses

Of the significant associations detected by our overall and stage analyses, we highlighted previously reported clinically relevant biomarkers, including immune cells, cytokines, and stromal cells (Figure 3D,E) [31]. Specifically, higher breast cancer PRSs were associated with lower regulatory T cells (Treg cells) [members of unassigned modules] (OR, 0.98; 95% CI, 0.97–0.9995; *p* = 0.0459) and lower CTLA-4 data [member of lymphocyte infiltration module] (0.76; 0.60–0.97; *p* = 0.0292) in advanced stages, and lower macrophage M2 (%) [member of unassigned module] (0.98; 0.96–0.99; 0.0038) and higher CD8^+^ T cells [members of T-cell/cytotoxic module] (1.002; 1.0003–1.004; *p* = 0.0221) in early stages. When we examined pro-tumoral cytokines, such as IL-2 and IL-4, we found that a high colorectal cancer PRS was associated with high IL2 score 21,050,467 (1.09; 1.004–1.18; *p* = 0.0434) and IL-4 score 21,050,467 (1.12; 1.03–1.21; *p* = 0.0116) levels in the early stages. Both immune traits were part of the lymphocyte infiltration module. A high colorectal cancer PRS was also associated with a high stromal fraction (%) [member of lymphocyte infiltration module] in the early stages (1.10; 1.02–1.17; *p* = 0.0097). Additionally, high prostate cancer PRSs were associated with an enhanced stromal fraction (%) (1.04; 1.01–1.07; *p* = 0.0194) and increased angiogenesis [members of the T-cell/cytotoxic module] (1.01; 1.0002–1.02; *p* = 0.0463).

### 3.6. Immune Subtype Association Analyses

We evaluated the association of PRSs with the six distinct immune subtypes of common cancers (Figure 3F). High breast cancer PRS was associated with a decreased C2 subtype (IFN-γ dominant) compared with the C1 subtype (wound healing) in advanced stages (OR, 0.24; 95% CI, 0.11–0.53; *p* = 0.0004). A high colorectal cancer PRS was significantly associated with a decreased C6 subtype (TGF-β-dominant) compared to the C1 subtype in the early stages, although wide CIs were observed owing to the small sample size. High lung adenocarcinoma PRSs were associated with a decreased C6 subtype in advanced stages, whereas high lung squamous cell carcinoma PRSs were associated with an increased C6 subtype in advanced stages, although wide CIs were observed owing to the sample size. The C5 subtype (immunologically quiet) was not detected in any type of cancer.

## 4. Discussion

To the best of our knowledge, this is the first study to systematically investigate the association between PRSs and immune traits in solid cancers. Among the 139 immune traits evaluated, we identified 31 significant associations among seven common cancers. Moreover, in the analyses stratified by stage, 65 associations were identified among the four common cancers. Notably, putative biomarkers, that could be effective in predicting antitumor immune responses in a clinical context, were identified. A low CTLA-4 expression was observed in the advanced stages of breast cancer.

Previous studies have shown that inherited genetic variants can affect host immune responses [21,32,33,34,35,36,37,38,39], and cancer-associated risk variants can modulate immune-related processes that can influence susceptibility to certain cancer types [40,41]. As an association-based study, our study has some limitations, including not showing robust mechanistic insight, not distinguishing between direct and indirect association, and not demonstrating physiological relevance or translational potential. However, our study results are consistent with those of several previous studies. For example, the association between PRSs and a limited number of traits, such as cell content, was studied in [9,23]. Consistent with our findings, Palomero et al. showed that the distribution of correlation coefficients between immune/stromal cell tissue content (neutrophils, gamma-delta T cells, CD8^+^ T cell activation, interferon response, Macrophage M2) and PRSs in breast cancer was less than zero. Conversely, positive correlations were principally detected in lung adenocarcinoma (gamma-delta T cells) and lung squamous cell carcinoma (cytotoxic cells, plasma cells, NK cells), although they were also observed in ovarian and prostate cancer [9]. However, these studies analyzed a limited number of traits and focused mainly on individual risk variants or genes. Interestingly, in our study, the most significant association was observed between PRS for prostate cancer and the immune traits. This could result from the large sample size and large number of SNPs. Additionally, chronic inflammation is prevalent in the adult prostate and likely plays a role in the formation of lesions that are putative risk factors for prostate cancer development [42].

We excluded tumors with microsatellite instability because they have fundamentally different biology and known anti-tumor immunity mechanisms. We also considered other known biological factors that influence antitumor immunity, such as the stage. Interestingly, in analyses stratified by stage, we found a greater number of significant associations between PRSs for cancers and immune traits than in overall association analyses. Previous studies have focused mainly on predicting prognosis according to the immune landscape in the early stages, owing to sample sizes [43,44,45,46]. Although this study had some sample size limitations, we systematically evaluated association analyses by early and advanced stages across the four cancer types. Furthermore, inflamed tumors are more responsive to anti-PD1/anti-PD-L1 therapy than those with large stromal infiltrations or an absence of T cells [35,47,48]. In the current study, CD8^+^ T cells represented inflamed tumors, whereas other Tregs, M2 macrophages, CTLA-4, IL-2, IL-4, stromal fraction, and angiogenesis represented non-inflamed tumors. We observed that high breast cancer PRS was associated with strong antitumoral traits in both early and advanced stages, whereas high colorectal cancer PRS was associated with strong pro-tumoral traits in the early stages. These observations may result from cancer heterogeneity and other potential unobserved characteristics. Further investigations are required to verify these putative biomarkers at different stages.

We observed that PRSs are significantly associated with immune subtypes, which may have clinical implications, such as prognoses and treatment outcomes. The C1 subtype was used as a reference [49]. Previous studies have suggested that IFN-signaling (C2) is associated with favorable prognosis and/or responsiveness to immunotherapy [50,51,52]. However, subtypes C4 and C6 have the worst prognoses and display signatures reflecting a macrophage-dominated, low lymphocytic infiltrate, with high M2 macrophage content, consistent with an immunosuppressed TME [30]. In this study, high breast cancer PRSs were associated with stronger pro-tumoral traits (decreased IFN-γ dominance) than the C1 subtype in advanced stages. High colorectal cancer PRSs were significantly associated with strong antitumor traits (decreased TGF-β dominance) in the early stages. High lung adenocarcinoma PRSs were associated with strong antitumoral traits (decreased TGF-β dominance) in advanced stages, whereas high PRS for lung squamous cell carcinoma was associated with strong pro-tumoral traits (increased TGF-β dominance) in advanced stages. Our results could aid the development of personalized treatment guidelines by identifying immune subtypes via PRSs.

This study systematically investigated the most recent GWAS-identified risk variants and used clinical characteristics, as well as genomic and transcriptomic data generated from over 2000 samples of common cancers from TCGA. However, there were limitations to this study. First, the number of cancer cases in TCGA was relatively small, and our associations did not reach the threshold after controlling for multiple comparisons (e.g., Bonferroni correction and false discovery rate). Notably, we observed that many of our association tests could reach the threshold at nominal *p* < 0.05, supporting an enrichment of true signals for our reported significant associations. In addition, we could not distinguish between stage III (locally advanced disease) and stage IV (advanced disease with distant metastasis) in stratified analyses because of the small sample size (number of stage IV cases: breast 12, colorectum 27, lung adenocarcinoma 15, lung squamous cell carcinoma 5, and pancreas, 4). Second, our observed associations between PRS and immune infiltration might not be directly related to the responses of immune cells to tumor signals. However, as PRS is derived from risk genetic variants associated with a cancer of interest, the observed association may be explained by potential dysregulated pathways in tumors. This was supported by previous genetic studies, wherein GWAS-identified risk variants played dysregulatory roles in cancer-related genes and pathways in target normal or tumor tissues [53]. Third, it should be noted that our analysis was based on the PRS constructed from common GWAS-identified risk variants. Other risk variants (i.e., rare) with epistatic effects, or expression of inflammatory genes (i.e., *COX1*, *COX2*, *LOX5* and *ALOX5AP* and correlated oncogenes), together with advanced approaches (e.g., interaction-aware [54] or machine learning-based PRSs), could be used to improve PRS-based analysis in future studies. Therefore, further molecular and cellular analyses are required to incorporate them and accurately assess their functional consequences with PRS. Fourth, some other factors might affect the extent or intensity of immune cell infiltration [55]. However, as the observed associations between PRS and immune cell infiltration are due to the different germline genetic factors among patients, other potential factors would not affect the overall effect of PRS on immune cell infiltration. Although it is still possible that some factors, such as therapeutic exposure, or time and duration of treatment, could be mediators for our observed associations, we were not able to investigate those associations due to the lack of treatment information in TCGA.

## 5. Conclusions

In conclusion, this study shows that germline features related to PRSs are associated with tumor immune infiltration in several common cancers. Our findings provide additional insights into the role of genetic susceptibility in immune responses, underlying cancer development, prognosis, and the potential role of early diagnostic or therapeutic targeting strategies.

## Figures and Tables

**Figure 1 cancers-14-05571-f001:**
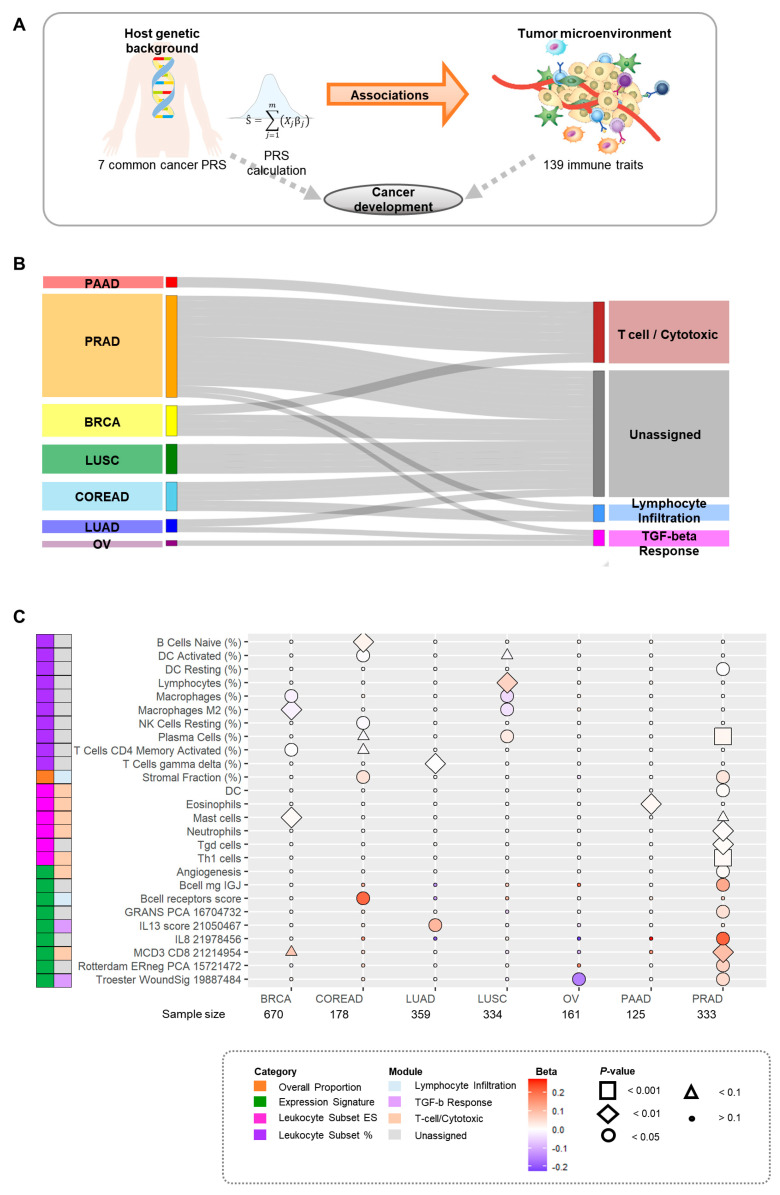
Overall associations between polygenic risk scores (PRSs) for cancers and immune traits. (**A**) Overall workflow for assessing the effect of PRSs for cancers on immune traits. (**B**) Overall proportion of cancer types in each immune module (nominal *p* < 0.05). The thickness of line indicates −log_10_
*P*. (**C**) Overall associations between PRSs for cancers and immune traits (nominal *p* < 0.05.) Immune traits were presented when a nominal *p* < 0.05 against at least one type of cancer was observed. The bottom figure represents the stratified analyses by intrinsic molecular subtypes of breast cancer between the PRSs for breast cancer and immune traits. Abbreviations: BRCA, breast cancer; COREAD, colorectal cancer; LUAD, lung adenocarcinoma; LUSC, lung squamous cell carcinoma; OV, ovarian cancer; PAAD, pancreatic cancer; PRAD, prostate cancer.

**Figure 2 cancers-14-05571-f002:**
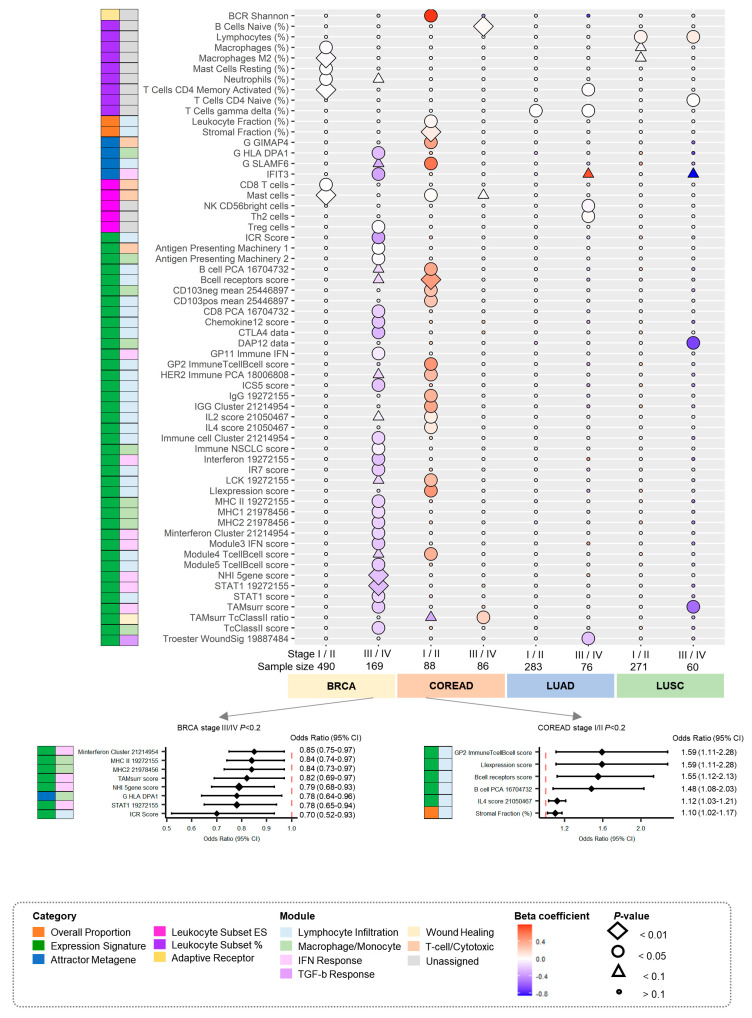
Stratified analyses by stage (early and advanced stages) between PRSs for cancers and immune traits. Immune traits were presented when a nominal *p* < 0.05. against at least one type of cancer, was observed. In the figure below, forest plots are presented when a nominal *p* < 0.02 in advanced stages of breast cancer and early stages of colorectal cancer, which were the most representative of the data. Abbreviations: BRCA, breast cancer; COREAD, colorectal cancer; LUAD, lung adenocarcinoma; LUSC, lung squamous cell carcinoma; PRS, polygenic risk score.

**Figure 3 cancers-14-05571-f003:**
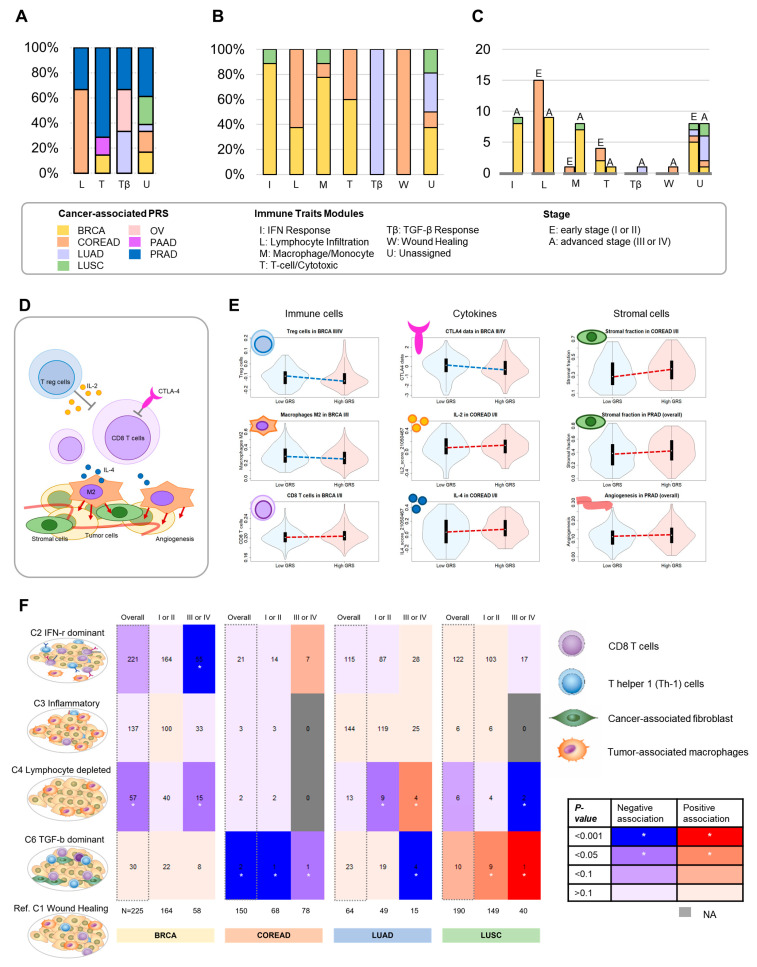
Summary of stratified analyses by stage and immune subtype. (**A**) Overall proportion of cancer types in each immune module (nominal *p* < 0.05). (**B**) Specific proportion by cancer stage in each immune module (nominal *p* < 0.05). (**C**) Absolute number of associations by cancer stage in each immune module (nominal *p* < 0.05). (**D**) Mechanisms of immune traits. Tregs attenuate the activity of CD4^+^ and CD8^+^ T cells to maintain self-tolerance through the secretion of immunosuppressive cytokines including IL-2. M2 macrophages are characterized by the expression of anti-inflammatory cytokines, such as IL-4 and chemokines, which suppress CD8^+^ T-cell activation, promote the recruitment of Tregs, and contribute to tumor immune evasion. CD8^+^ T cells play a central role in inducing an antitumor immune response through the release of cytolytic factors and induction of apoptosis in tumor cells. CTLA-4 on T cells leads to a pro-tumoral immunosuppressive phenotype. Stromal components can contribute to immune evasion and resistance to immune checkpoint inhibitors. High rates of angiogenesis in the tumor microenvironment, resultant abnormal vasculature, and high interstitial pressure within the tumor can impair the infiltration of immune cells and penetration of checkpoint inhibitors. (**E**) Representative immune traits by PRS in the cancer groups. The PRS group was classified into “low PRS” and “high PRS” based on the median PRS for each cancer. White dots represent the median values. The thick black bars in the center represent interquartile ranges. The thin gray line represents the rest of the distribution, except for the points that were determined to be outliers. (**F**) Immune subtype analyses. The numbers in the matrix indicate the sample size for each group. Abbreviations: BRCA, breast cancer; COREAD, colorectal cancer; CTLA-4, cytotoxic T-lymphocyte-associated protein 4; IL, interleukin; LUAD, lung adenocarcinoma; LUSC, lung squamous cell carcinoma; OV, ovarian cancer; PAAD, pancreatic cancer; PRAD, prostate cancer; PRS, polygenic risk score; Treg, regulatory T cells.

**Table 1 cancers-14-05571-t001:** Genetic variants used in association analyses with cancer risk for each of the common cancers.

Cancer Type ^a^	Abbreviation	SNPs ^b^	Samples ^c^
Breast cancer	BRCA	324	670 (25.2%)
Colorectal adenocarcinoma	COREAD	129	178 (48.3%)
Lung adenocarcinoma	LUAD	19	359 (21.2%)
Lung squamous cell carcinoma	LUSC	17	334 (18.0%)
Ovarian serous cystadenocarcinoma	OV	26	161 (—)
Pancreatic adenocarcinoma	PAAD	21	125 (6.4%)
Prostate adenocarcinoma	PRAD	257	333 (—)

Abbreviations: PC, principal component; SNP, single nucleotide polymorphism; TCGA, Cancer Genome Atlas. ^a^ We used regression models adjusted for histological subtype and stage, sequencing center, and the top five PCs for genetic ancestry for BRCA; histological subtype and stage, sex, sequencing center, and the top five PCs for genetic ancestry for COREAD; stage, sex, sequencing center, smoking history, and the top five PCs for genetic ancestry for LUAD and LUSC; sequencing center and the top five PCs for genetic ancestry for OV and PRAD; stage, sex, sequencing center, and top five PCs for genetic ancestry for PAAD. ^b^ Corresponding to the number of SNPs after removing SNPs that were unavailable in TCGA datasets. ^c^ All samples included microsatellite instability-stable tumors. Parentheses indicate the proportion of patients with stage III or IV disease.

## Data Availability

Data supporting the findings of this study are available from the corresponding author upon reasonable request.

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
