# Peer review of "Polygenic Risk Scores Associated with Tumor Immune Infiltration in Common Cancers"

_cancers, 2022, doi:10.3390/cancers14225571_

Round 1

Reviewer 1 Report (Previous Reviewer 1)

In this study, Choi et al. evaluate the association between polygenic risk scores (PRS) with immune cell infiltration in common solid tumors. They studied the common genetic risk factors in different tumors, with 139 previously defined immune traits. They identified 31 significant associations between PRS and immune traits. They also observed significant associations when stratifying the dataset by stage of cancer.

The study is comprehensive and well-written, which certainly has future implications for early diagnosis of cancers. However, the authors should address the following concerns for the article to be acceptable:

  1. The study is primarily associated-based. Although such studies certainly are essential, they have several limitations. These include a lack of robust mechanistic insight, direct vs. indirect association, physiological relevance, translational potential, etc. Furthermore, the subject size is relatively smaller in this study. At the least, the authors need to add a section/paragraph to discuss these caveats and cite relevant literature to associate their findings with the existing literature showing similar phenotypes.
  2. Immune cell activity is a complex response to a set of autocrine and paracrine signals. Although the authors observe associations with PRS and immune infiltration, it might not be directly related to how the immune cells respond to tumor signals. Are the PRS indicative of issues within the immune cells? Or do the PRS primarily relate to tumors and how the tumor-associated signals are transmitted and later perceived by the immune system?
  3. Some critical factors when studying cancers apart from their stage are the duration of cancer, age of the subjects, therapeutic exposure, and time/duration of treatment. These factors also define the extent or strength of immune cell infiltration. Is there an association of PRS with these factors?

Author Response

Reviewer 2 Report (Previous Reviewer 2)

The authors have addressed concerns of the reviewers and the manuscript is improved.  Nevertheless, they need to acknowledge that other non-mutated genes have been noted to have considerable impact on carcinogenesis, e.g, in particular expression of inflammatory genes, COX1, COX2, LOX5 and ALOX5AP and correlated oncogenes.

Round 2

Reviewer 1 Report (Previous Reviewer 1)

The authors have addressed most of my concerns. This manuscript can be accepted in its present form.

This manuscript is a resubmission of an earlier submission. The following is a list of the peer review reports and author responses from that submission.

Round 1

Reviewer 1 Report

In this article, Choi et al. investigated the association between polygenic risk scores (PRS) and tumor immune infiltration in different cancers. Upon analysis, they identified significant associations between PRS and immune traits in prostate cancer, colorectal cancer, breast cancer, lung squamous cell carcinoma, lung adenocarcinoma, ovarian cancer, and pancreatic cancer. This study is comprehensive and novel. It clearly shows the genetic risk component in the immune cell infiltration of tumors. Although this study is interesting, there are a few concerns that the authors need to address:

1. The article has excellent findings, but it needs to be better organized entirely. At this stage, the article is filled with technical jargon without a proper flow of the text. Especially, the methods need to be explained well. The authors seem to have taken data from different sources and combined it in some cases. So, it should be appropriately explained. Overall, the format of the article is completely based on the assumption that the reader will understand the intricacies of the data collection, analysis, and interpretation.

2. The authors have to explain the figures in depth. So far, the authors only have mentioned the big picture outcome of the figure. For every figure, they need to explain the axes, color code, and clustering of the data if possible. Adding a plethora of data without explanation makes it difficult to assess the interpretation. For e.g., In figure 2, there is a mention of “Model5 T cell B cell score” or “ICR score,” etc. The authors need to detail such data and explain the findings for all figures.

3. The authors speak about the analysis of immune traits. What immune traits were studied?

4. Which were the major polygenic risk variants? Were there any common risk variants among all cancers?

5. The in-depth analysis of breast cancers, as opposed to other cancers, is clouding the message of the article.

5. The authors need a better conclusion/discussion to demonstrate the importance of these findings and future applications. Can this study be utilized for potential early diagnostic or a therapeutic targeting strategy?

Reviewer 2 Report

The report attempts to justify an association between an index of genetic mutations and the expression of genes related to immunity.  The regression analysis is unclear and the measures of effects are not quantified, e.g., how exactly were the effects quantified and what is the distribution of these effects?  Were there specific genetic mutations that had a relatively greater impact on immune function than others?  If so, what was the nature of such associations and how were they related to carcinogenesis?  How were genotype x environment interactions handled in the analysis?  Were there other independent variables included in the model to account for the possibility of such interactions?   

Reviewer 3 Report

In this work, the authors constructed polygenic risk score (PRS) for some common cancers in order to systematically evaluate the association between these PRSs and immune traits for each of the tumors considered.The proposed paper is a useful and original topic that makes it interesting and worthy of consideration.It is easy to understand because it is well constructed, clear and well described with comprehensive figures appropriate to the subject matter. The scientific background and objectives are clearly explained and seem appropriate to the field of investigation. The information gathered in this work could have important implications for a better understanding of the role of genetic susceptibility in immune responses underlying cancer development and prognosis. The topic falls within the subject area of the journal.

In my opinion, this work is acceptable for publication in Cancers after minor revisions that will help the authors to improve the quality of their manuscript.

- In the introduction please cite and discute the following recent articles about the role of tumour-infiltrating lymphocytes in cancer (PMID: 35095898; https://doi.org/10.3390/cancers14184344; DOI: 10.3389/fonc.2022.946319; doi:  10.1186/s40425-016-0165-6);

- Based on your knowledge and data in the literature, how do you explain that, among the various common cancer types analysed, the most significant association was observed between the PRS for prostate cancer and immune traits?

- Please carefully revise the English language throughout the text and correct some grammatical errror and trivial imperfections.